# Differences in the Molecular Profile between Primary Breast Carcinomas and Their Cutaneous Metastases

**DOI:** 10.3390/cancers14051151

**Published:** 2022-02-23

**Authors:** Silvia González-Martínez, David Pizarro, Belén Pérez-Mies, Tamara Caniego-Casas, José Luis Rodríguez-Peralto, Giuseppe Curigliano, Alfonso Cortés, María Gión, Javier Cortés, José Palacios

**Affiliations:** 1Clinical Research, Ramón y Cajal Hospital, 28034 Madrid, Spain; silviagonzalezmartinezbio@gmail.com; 2“Contigo Contra el Cáncer de la Mujer” Foundation, 28010 Madrid, Spain; 3Molecular Pathology of Cancer Group, Ramón y Cajal Health Research Institute (IRYCIS), 28034 Madrid, Spain; david.pizarro@salud.madrid.org (D.P.); bperezm@salud.madrid.org (B.P.-M.); tamara880723@hotmail.com (T.C.-C.); 4Department of Pathology, Ramón y Cajal University Hospital, 28034 Madrid, Spain; 5Centre for Biomedical Research in Cancer Networks (CIBERONC), Carlos III Health Institute, 28029 Madrid, Spain; jrperalto@salud.madrid.org; 6Faculty of Medicine, University of Alcalá, 28801 Madrid, Spain; 7I+D Institute, 12 de Octubre University Hospital, 28041 Madrid, Spain; 8Department of Pathology, Medical School, Complutense University, 28040 Madrid, Spain; 9European Institute of Oncology, Scientific Institute for Research, Hospitalization and Healthcare (IRCCS), 20141 Milan, Italy; giuseppe.curigliano@ieo.it; 10Department of Oncology and Hematology, University of Milan, 20122 Milan, Italy; 11Department of Medical Oncology, Ramón y Cajal University Hospital, 28034 Madrid, Spain; acsalgado86@gmail.com (A.C.); mariagion@gmail.com (M.G.); 12Department of Medicine, Faculty of Biomedical and Health Sciences, European University of Madrid, 28670 Madrid, Spain; 13International Breast Cancer Center (IBCC), Quironsalud Group, 08017 Barcelona, Spain; 14Medica Scientia Innovation Research, 08007 Barcelona, Spain; 15Medica Scientia Innovation Research, Ridgewood, NJ 07450, USA

**Keywords:** breast cancer, metastasis, skin, NGS, mutations, pathology, immunohistochemistry

## Abstract

**Simple Summary:**

The development of new strategies for the management of cutaneous metastases is a major clinical challenge because of the poor prognosis. To advance in this field, a better understanding of the molecular alterations involved in the metastatic process is needed. In the present study, the clinicopathological characteristics of breast cancer that develop cutaneous metastases were analyzed and the molecular differences between primary breast tumors and their corresponding cutaneous metastases were compared. We observed that the surrogate molecular type of breast cancer with an increased risk to metastasize to the skin was triple negative. In total, 48.5% of the cutaneous metastases presented some additional molecular alteration with respect to the primary tumor. However, no characteristic mutational pattern related to skin metastasis development was observed. Identifying the genes involved in the development of cutaneous metastases is important to gain insights into the biology of the disease and to identify possible diagnostic and therapeutic biomarkers.

**Abstract:**

Background: The characterization of molecular alterations of primary breast carcinomas (BC) and their cutaneous metastases (CM) to identify genes involved in the metastatic process have not yet been completely accomplished. Methods: To investigate the molecular alterations of BC and their CM, a total of 66 samples (33 BC and 33 CM) from 33 patients were analyzed by immunohistochemical and massive parallel sequencing analyses. In addition, the clinicopathological characteristics of patients and tumors were analyzed. Results: Triple negative (TN) BCs were overrepresented (36.4%) among tumors that developed CM. A change of tumor surrogate molecular type in metastases was found in 15% of patients and 48.5% of the CM presented some additional molecular alteration with respect to the primary tumor, the most frequent were amplification of *MYC* and *MDM4*, and mutations in *TP53* and *PIK3CA*. Survival was related to histological grade, tumor surrogate molecular type and *TP53* mutations in the univariate analysis but only the tumor surrogate molecular type remained as a prognostic factor in the multivariate analysis. Conclusions: The TN molecular type has a greater risk of developing skin metastases. There are phenotypic changes and additional molecular alterations in skin metastases compared to the corresponding primary breast tumors in nearly half of the patients. Although these changes do not follow a specific pattern and varied from patient to patient, they could impact on the treatment. More studies with larger patient and sample cohorts are needed.

## 1. Introduction

Breast cancer (BC) is the most prevalent malignancy in females and is the leading cause of cancer death in women [1]. We can distinguish between different groups of BC according to the molecular profile as those that express Estrogen Receptor (ER) and/or Progesterone Receptor (PR) (75%), those that express Epidermal Growth Factor Receptor 2 (Her2) (15%) [2], and tumors that do not express any of these three markers, the triple-negative (TN) tumors (10–20%) [3]. According to these characteristics, we can apply a surrogate molecular classification that distinguishes four surrogate molecular types of BC: luminal A-like, luminal B-like (HER2− and HER2+), HER2+ (non-luminal), and triple negative (TN) [4].

Metastasis accounts for the majority of deaths from BC [5]. It is a complex process in which the cells of a primary tumor are propagated to distal organs, showing uncontrolled growth in these tissues [6,7]. Normally, BC metastasize to the lung, bone, and brain [8]. Moreover, BC is the tumor most prone to develop skin metastases in females [9].

Skin metastases are the result of lymphatic embolization, hematogenous or contiguous spread [10] and are present in around 24% of patients with metastatic BC [9,11,12,13,14,15,16]. In addition, due to the high incidence of BC, these skin manifestations are the most common metastases among women seen by dermatologists, specifically, 69% of these metastases come from BC [13].

Cutaneous breast metastases tend to develop in the vicinity of the primary tumor in the skin of the breast and chest wall, although they can also develop in the abdomen, extremities, head, or neck [9,12,13]. In addition to being able to develop in different locations, breast cutaneous metastases (CM) can manifest in a wide variety of ways. Nodules are the most common presentation (80%), but there are other patterns [11,13] such as telangiectatic pattern with an incidence of 8–11%, erectile pattern with an incidence of 3–6.3%, carcinoma en cuirasse with an incidence of 3–4% and neoplastic alopecia with an incidence of 2–12% [11,13].

All BC metastases, regardless of subtype, usually occur late in the disease, in the later stages of the disease course. Brownstein et al. [17] observed that skin metastasis was the presenting sign of the disease in only 3% of cases of metastatic BC. Kong et al. [15] observed that 56.8% of the patients had more than one visceral metastasis at the time of diagnosis of CM. Therefore, when diagnosed, the primary tumor is widespread and may not be treatable (palliative care, surgical excision, or complete mastectomy is provided) [18]. All this confers a poor prognosis, with an average survival of 3 to 6 months, with few differences regarding whether the lesions are single or multiple, with mortality exceeding 70% in the first year after diagnosis [19].

Since there are few series evaluating the molecular alterations of CM and most do not include the analysis of both the primary tumor and CM, the objective of this study was to compare the molecular alterations of matched primary BC and their CM in a series of 33 patients to better understand the genes implicated in BC progression and to identify potential therapeutic targets.

## 2. Materials and Methods

Compliance with Ethical Standards: All procedures performed in studies involving human participants were in accordance with the ethical standards of the institutional and national research committee and the 1964 Helsinki declaration and its later amendments or comparable ethical standards. The use of patient samples to meet the objectives of this study was approved on 14 May 2021 by the Ethics Committee of the Hospital Ramón y Cajal (ethical approval code: 30-21).

### 2.1. Histology

Histological sections of all primary tumors and their metastases were reviewed by two experienced pathologists (J.P and B.P.-M.). Histological typing and grading was performed according to WHO recommendations [4]. Lymphovascular invasion (LIV) was also evaluated in primary tumors. Cutaneous lesions were diagnosed as metastases from BC based on a biopsy, excluding cases with direct extension from a subjacent breast lesion.

### 2.2. Immunohistochemistry

All 33 primary tumors and 33 metastases underwent an immunohistochemical study for the expression of estrogen receptors (ER), progesterone receptors (PR), HER2 and Ki67. Immunostaining was performed using the EnVision detection system (K5007, Agilent Dako, Glostrup, Denmark) using the following antibodies: ER (clone EP1, Agilent Dako Omnis, Glostrup, Denmark), PR (clone PR 1294, Agilent Dako Omnis, Glostrup, Denmark), HER2 (SK001, clone poly, Agilent Dako autostainer, Glostrup, Denmark), and Ki-67 (clone MIB-1, Agilent Dako Omnis, Glostrup, Denmark). Evaluation of ER, PR, and HER2 expression was performed according to American Society of Clinical Oncology and the College of American Pathologists (ASCO-CAP) guidelines) [20]. HER2 equivocal cases (2+) underwent FISH analysis, using the PathVysion HER-2 DNA Probe Kit (PathVysion II kit, Abbot Laboratories, Abbot Park, IL) on complete tumor sections. Results were interpreted according to 2018 ASCO-CAP guidelines [20].

In the 7 invasive lobular carcinomas, the expression of E-cadherin (clone NHC-38, Agilent Dako Omnis, Glostrup, Denmark) was studied to confirm the histological type. Androgen receptor (AR) (SP107 Cell Marque, Ventana Medical Systems, Rocklin, CA, USA) was also determined to evaluate the possible apocrine phenotype in 12 primary TN tumors and their metastases.

Tumors were classified into different surrogate molecular types as Luminal HER2-, Luminal HER2+, HER2 (non-luminal), and TN.

### 2.3. Massive Parallel Sequencing

Sufficient DNA for sequencing was obtained from the 33 primary breast carcinomas and their corresponding CMs. Areas with >30% of tumor cells were obtained by “punching” paraffin blocks in selected areas previously marked on hematoxylin/eosin (H&E) slides. The QIAamp DNA FFPE Tissue Kit (Qiagen, Valencia, CA, USA) was used to extract DNA from all samples. Quality of DNA samples was measured using TapeStation (Agilent 2200 TapeStation, Santa Clara, CA, USA), whereas quantification was performed by QUBIT 2.0. (Thermo Fisher Scientific Qubit 2.0 Fluorometer, Waltham, MA, USA).

A custom gene panel was designed using the SureDesign platform by Agilent Tech. (Santa Clara, CA, USA) to consistently target 61 genes (*AKT1, ARID1A, ARID1B, ARID5B, ATR, BCOR, BRCA1, BRCA2, BRAF, CASP8, CCNE1, CDH1, CDH4, CDH19, COL1A1, CSMD3, CTCF, CTNNB1, EGFR, ERBB2, ESR1, FBXW7, FGFR1, FGFR2, FOXA2, GRB7, GSDMB, MAP2K4, KRAS, MAP3K1, MLL3, MLH1, MKI67, MSH2, MSH6, MYC, NCOR1, NF1, NRAS, PGAP3, PIK3CA3, PIK3R1, PMS2, PNMT, POLE, PPP2R1A, PRPF18, PTEN, KMT2B, RB1, RPL22, SF3B1, SPOP, STARD3, TAF1, TBX3, TCAP, TP53, VGLL1, ZNF217, ZNF703*). For library construction, a modified protocol for Agilent SureSelect^XT^ FFPE was selected based on Covaris AFA fragmentation of DNA (Covaris, Woburn, MA, USA) and subsequent probe-mediated hybridization capture. Sequencing of equimolar libraries was performed using the Ion S5™ Torrent (Thermo Fisher Scientific, MA, USA).

Bioinformatics analysis was carried out using a specific pipeline using Novoalign V3 (2021) (http://www.novocraft.com/products/novoalign/ accessed on 4 February 2022) as aligner and VarScan [21] as variant-caller, with no filters. Variant annotation was performed using the VEP from Ensembl version 88 (http://www.ensembl.org/info/docs/tools/vep/index.html accessed on 4 February 2022), which corresponds to the hg38 version of the human reference genome. Variants were latterly filtered using the functional information (taking only deleterious variants), the variant allele frequency (>0.05), and the strand-bias from both the variant and the reference allele. If normal tissue was available, those variants also present in the normal component were excluded. Finally, visual inspection was performed as the final selection criterion using the IGV browser [22].

In addition, 11 mutations were confirmed by Sanger sequencing, 3 in the *PIK3CA* gene (Pt3, Pt14 and pt36), 6 in the *TP53* gene (Pt1, Pt2, Pt4, Pt10 and Pt11), and 2 in the *ERBB2* gene in the samples (pt13). (Appendix A).

### 2.4. Fluorescent In-Situ Hybridization on Tissue Microarrays

Since our panel was not designed to detect CNVs, a tissue microarray (TMA) was constructed to evaluate gene copy number variations in *CCND1, MYC, FGFR1* and *MDM4,* the genes most frequently amplified in BC, by Fluorescent In-Situ Hybridization (FISH). Only 20 matched primary tumors and metastases (40 samples) were included in the TMA due to sample limitation after the initial immunohistochemical/molecular study. Chromosomal alterations were evaluated by FISH on TMA sections using the following probes: SPEC *CCND1/CEN11*, *MYC/CEN8, FGFR1/CEN8*, and *MDM4*/1p12 dual color Probe Kit (Zytovision GmbH, Bremen, DE). FISH slides were observed with a fluorescence microscope at 100X with immersion oil. A detailed scoring of at least 20 neoplastic cells per sample was performed. Amplification was considered when the tumor cell population had at least twice as many gene signals than centromere signals of the respective chromosome (ratio ≥ 2), and polysomy when the average of centromere signals on tumor cells were >3.

### 2.5. Statistical Methods

The Kaplan–Meier method was used to calculate overall survival according to clinicopathological characteristics (age, pT, pN, histological type, surrogated molecular type, LIV, histological grade, clinical stage, metastasis location, and neoadjuvant therapy) and mutations in *TP53* and *PIK3CA* genes. Cox proportional hazards models were used to investigate the association between mortality and clinicopathological and molecular features.

## 3. Results

### 3.1. Case Selection

A total of 33 patients diagnosed with BC and CM between 2005 and 2020 from the Pathology Department in Ramón y Cajal University Hospital (Madrid, Spain) and in 12 de Octubre University Hospital (Madrid, Spain) were selected, all had available paired samples (primary tumor and CM).

### 3.2. Clinicopathological Features

Clinicopathological features of all 33 primary samples are presented in Appendix A and summarized in Table 1. The median age of the patients at diagnosis was 63.5 years old (range 29 to 84), 57.6% of patients were older than 60 years.

According to the immunohistochemical profile, 16 cases (48.5%) were Luminal HER2- surrogate molecular type, 3 cases (9%) were Luminal HER2+, 2 cases (6%) were HER2+ (non-Luminal), and 12 cases (36.4%) were TN.

Examining the distribution of histological types, 24 cases (72.7%) corresponded to invasive breast carcinomas of non-special type (BCNST), the rest were special histological types. The largest group of 7 cases (21.2%) corresponded to invasive lobular carcinoma (ILC), one case (3%) was a matrix-producing (chondroid) metaplastic carcinoma, and one case was an apocrine carcinoma (3%).

### 3.3. Immunohistochemistry and HER2 FISH

#### 3.3.1. Surrogate Molecular Types

The surrogate molecular type of the tumors and metastases was confirmed by IHC and FISH. The most frequent type of primary tumor was Luminal HER2- followed by TN. The molecular type changed between the primary tumor and its CM in 5 patients (15%), the most common being from luminal to TN (Table 2).

In case Pt14, HER2 amplification was lost in the metastasis; on the other hand, in case Pt32, HER2 was overexpressed in the metastasis due to polysomy of chromosome 17.

#### 3.3.2. Androgen Receptor in Triple-Negative Cases

AR expression was studied in 12 TN primary tumors and their metastases by IHC, since AR expression in TNBC is related with the apocrine molecular type [23]. In this series, 3 out 12 TNBC expressed AR and expression was concordant in primary tumors and the associated metastases.

### 3.4. Molecular Characterization

#### 3.4.1. Most Frequently Altered Genes in Primary Tumors and Cutaneous Metastases

The following molecular analysis is based on 66 samples (33 primary tumor and 33 CM) for mutation analysis and 40 samples (20 primary tumor and 20 CM) for CNV analysis. Figure 1 shows the mutations and CNVs found in 29 pairs. In 4 cases, no mutations or CNVs were found, and all were Luminal HER2- cases. Molecular alterations were detected in 12 TN, 12 Luminal HER2-, 3 Luminal HER2+ and 2 HER2+ (non-Luminal) primary tumors. The number of mutations ranged between 1–5. A summary table with the type of mutation and CNVs found in each of the samples is presented in Appendix A.

Among the 33 matched cases, *TP53* was mutated in 13 primary tumors (39.4%) and in 14 CMs (42.4%). *PIK3CA* was mutated in 13 primary tumors (39.4%) and 15 CMs (45.4%). *NF1* was mutated in 3 of the primary tumors (9%) and 4 CMs (12.1%). *AKT1* was mutated in 3 primary tumors (9%) and 3 CMs (9%). *ERBB2* was amplified or there was polysomy in 5 primary tumors (15.15%) and 5 CM (15.15%). Among 20 matched cases, *MYC* was amplified in 3 primary tumors (15%) and 5 CM (25%). *MDM4* was amplified in only 3 CM (15%). *FGFR1* CNVs (amplification or polysomy) were observed in 2 primary tumors (10%) and 2 CMs (10%). Finally, *CCND1* was amplified in 1 primary tumor (5%) and 1 CM (5%). Table 3 shows a summary of these alterations distributed by surrogate molecular types.

Analyzing *TP53* and *PIK3CA* mutation frequencies in the CMs of BC cases diagnosed at early stages (I-II) versus those diagnosed at advanced stages (III-IV), we observed that they were very similar. The mutation frequency for both genes of cases diagnosed at early stages was 44.4%. In cases diagnosed at advanced stages the frequencies were 47.8% for *TP53* and 52.2% for *PIK3CA*. Thus, there were no statistically significant differences between the two groups.

##### Enriched Molecular Alterations in Cutaneous Metastases

Additional molecular alterations were observed in the CM, either mutations or CNVs, in 16 patients (48.5%) (Figure 2). In 7 cases (21.2%) there was more than one additional alteration in the CM. Table 4 shows the distribution of these alterations by surrogate molecular types.

There were some differences in the frequency of additional mutations depending on the local or distant nature of the CM. In the 17 paired cases that developed distant CM, additional alterations in the CM with respect to the primary tumors were found in 11 cases (64.7%) (5 TN, 4 Luminal HER2-, and 2 HER2+). In contrast, among the 15 paired cases that developed local CM, only 5 cases (33.3%) (3 TN, 1 Luminal HER-, and 1 HER2+) showed additional molecular alterations. (Table 4).

In addition, 11 mutations were confirmed in the *TP53, PIK3CA*, and *ERBB2* genes. Figure 3 shows the different mutations found in the *ERBB2* gene (L755S and S310F) between the primary tumor and the MC of Pt13, as well as the verification by Sanger.

### 3.5. Survival Analysis

In our series, 7 patients were alive and 26 had died when the data were censored. The median survival since the diagnosis of the disease was 53 months and the median survival since the diagnosis of the CM was 19.6 months. Of the patients who developed distant CM, the median survival was 14.5 months, with an overall survival in the first year of 46.7% of patients. In contrast, when MC was local, the median survival was 34.6 months, with an overall survival in the first year of 40% of patients.

The association between overall survival since the diagnosis of the primary tumor and age, location (local or distant), pT, pN, clinical stage, surrogate molecular type and histological type, histological grade, LIV, and neoadjuvant therapy were assessed. In addition, the association between overall survival and *TP53* and *PIK3CA* mutations was assessed. Histological grade, surrogate molecular type, and *TP53* mutations significantly affect overall survival (*p* = 0.015, *p* = 0.0011 and *p* = 0.019, respectively) (Figure 4a–c). The shortest overall survival was observed in the TN surrogate molecular type (Figure 4a). By multivariate analysis with the 3 significant variables, the only independent variable in the CoxPh analysis was surrogate molecular type, where HER2- Luminal was the best prognostic type. (Figure 4d).

In contrast, the variables pT, pN, stage, permeation, age, histological type location (local or distant), neoadjuvant therapy, or *PIK3CA* mutations did not significantly affect overall survival.

## 4. Discussion

### 4.1. Clinicopathologic Features of Breast Carcinomas That Develop Cutaneous Relapses

In this study, the clinicopathological features were analyzed of 33 females with BC that developed cutaneous metastases, distantly in 53.1% and locally in 46.9% of the patients. Our observation that 46.9% of CM developed locally on the skin of the breast/thorax is in accordance with the review by Johnson et al. [24], who found in a study of 61 patients with CM, 57% showed metastases in the breast/thorax skin.

In the present series, CM developed at a median of 22.8 months after the initial diagnosis of the primary BC. CM usually appear at the end of the disease, late during cancer evolution. Thus, Lookingbill et al. [25] found that only 6.3% of patients with BC had cutaneous involvement at the time of diagnosis of the primary tumor. More recently, Johnson et al. [24] reported that approximately 13.7% of the patients in the 9 retrospective series reviewed [26,27,28,29,30,31,32,33,34] had a skin lesion before or simultaneously with the diagnosis of BC. Supporting these observations, we found that 18.2% of patient in the present series debuted with skin involvement.

The median age of women with BC who developed CM varies among series. Whereas the age in our series (63.5 years) was similar to the Johnson et al. [24] review of 41 patients, other studies have reported a mean of 74 years (*n* = 18) [9] and 48 years (*n* = 125) [15]. These differences may be related with the different number of patients included in each series or due to ethic/geographic differences, since these series originated from Italy, South Korea, and Spain.

In the present series, the histology of 24 cases (72.7%) corresponded to BCNST, and the rest to special histologic subtypes. Among them, the largest group corresponded to ILC (21.2%). However, considering only the 16 Luminal HER2- cases in our series, 43.7% were ILC. Since the frequency of ILC in Luminal HER2- BC is between 15% and 20% [35], our results suggested that ILC is a major risk factor for developing skin metastases. In accordance with our results, the study of Li et al. [36] that included only metastases from Luminal HER2- BC, reported that the proportion of ILC was 26% in the group of patients that developed CM.

We observed that the frequency of BC developing CM varied according to the subtype. Table 5 compares the data from our series with those from other published series [15,37,38,39], including our recently published CM review that includes samples (*n =* 58) of molecularly characterized CM [39,40,41,42,43].

Although there were differences between series, a finding common to all of them is the relative overrepresentation of TNBC, since its frequency in the general population of BC is around 15% [44], but it was between 23–39% in BC with CM, suggesting that this surrogate molecular type could be associated with a greater potential to metastasize to the skin.

In our series, 15.2% of BC that developed CM were HER2+, a frequency similar to the general population of BC, at least in Spain, suggesting that this surrogate molecular type does not have a special propensity to develop CM. However, Table 5 shows important differences in the frequency of HER2+ BC that develop CM among different series. These differences are probably due to differences in sample size, patient selection, methods of HER2 detection and the period of study.

### 4.2. Molecular Alterations Involved in the Development of Cutaneous Metastasis in Breast Cancer

In this series, we studied the mutational landscape of 33 matched primary tumors with their corresponding CM by NGS. In addition, changes in copy number of *CCND1, FGFR1, MDM4* and *MYC* were analyzed by FISH in 20 paired samples. The molecular landscape of primary tumors in this series was concordant with many previous reports demonstrating different mutational patterns among different surrogate molecular types. Thus, whereas mutations in *PIK3CA* predominated among Luminal BC, *TP53* mutation was the main molecular alteration in TNBC.

When primary tumors were compared with their respective metastasis, we found that 48.5% of CM exhibited additional pathogenic mutations and/or gene amplification in important oncogenes and/or tumor suppressor genes. Genes involved in the progression of more than one case in the present series included *MYC, MDM4, PIK3CA* and *ERBB2.* In spite of this high frequency of additional changes, we did not observe a specific mutational pattern related to tumor progression, indicating that CM is a very individual process in each tumor. These results add to the observations of three previous series analyzing paired primary BCs and their CMs including a total of only 15 patients [39] (Table 6).

There are several studies analyzing molecular alterations in metastatic BC including samples from CM, but without comparison with their primary tumors (see González- Martínez et al. [37]). In general, these studies also confirmed that there is no specific pattern of mutations that predispose to CM. Only Rinaldi et al. [39] observed that alterations in *NOTCH1* were overrepresented in CM when compared to other metastatic locations.

We observed that the number of CM with additional molecular alterations was higher in distant than local metastases. Thus, additional mutations and CNVs were observed in 11 out of 17 (64.7%) distant CM but in only 5 out 15 (33.3%) local CM. These differences seemed to indicate a more advanced molecular stage of distant CM, although they did not have prognostic implications.

It is also worth noting that there was hardly any difference between the mutation frequency of *TP53* and *PIK3CA* in CM samples of patients diagnosed at early and advanced stages, so we assume that treatment has no effect on the mutation pattern.

### 4.3. Therapeutic Implications

An important question in the study of cancer metastasis is whether or not the molecular characteristics of metastatic samples modify patient treatment. In 5 patients in this study (15.1%), there was a conversion of the surrogate molecular type between the primary tumor and its CM. In 3 cases (9%), there was a change from luminal to TN, from Luminal HER2+ to Luminal HER2- in one case (3%), and from Luminal HER2- to Luminal HER2+ in another case (3%). All these changes would have impacted on the treatment decision regarding the use of hormone therapy or anti-HER2 therapy.

Regarding specific mutations, three CM developed additional *PIK3CA* mutations, which is a target for treatment with alpelisib in RH+ metastatic BC. However, the three primary and metastases were TN. We detected an *ESR1* mutation in a primary HR+ tumor and the corresponding CM. *ESR1* mutations are the most frequent additional mutations that develop in metastatic HR+ BC after hormone therapy, especially after the use of aromatase inhibitors, being infrequent in primary tumors. In this case, *ESR1* mutations would have influenced the response to hormone therapy during the complete evolution of the disease.

An interesting case in this series was the tumor and the CM developed in patient Pt13, which was an ILC that carried the pathogenic L755S mutation in the primary tumor. This mutation was absent in the CM, which carried the pathogenic mutation S310F. *ERBB2* mutations are more frequent in ILC (6%) than in IDC (1.5%), especially in ILC with pleomorphic features, and are associated with a poor prognosis in ILC [45,46]. In addition, response to different anti-HER2 therapies differed among mutations. Thus, whereas L755S seems to be resistant to trastuzumab and lapatinib but sensitive to neratinib and afatinib, S310F seems to be sensitive to all drugs.

### 4.4. Prognosis

Patients with CM have a very poor prognosis. Lookingbill et al. [47] observed an average survival of 31 months after the diagnosis of the CM. Kong et al. [15] observed a median survival of 32 months. In our series, the median survival was 19.6 months, and this difference may be related to the clinical differences between the series. Additionally, patients who developed distant CM had a shorter overall survival and died more frequently during the first year after the diagnosis of CM. However, these differences between survival according to location were not statistically significant. In the univariate survival analysis, the only three variables that showed an impact on prognosis were the histological grade, the surrogate molecular type, and *TP53* mutations. However, in the multivariate analysis, only the surrogate molecular type remained statistically significant, and the Luminal HER2- surrogate molecular type had a better prognosis.

### 4.5. Study Limitations

One limitation of this study is the relatively low number of cases studied. However, in spite of this and to the best of our knowledge, this is the largest series analyzing matched primary tumors and CMs reported so far. In addition, a comprehensive NGS panel of 61 genes was used in this study, including the most frequently mutated genes in BC. However, there may be additional genes with a role in progression in individual tumors that were not included in the panel. Moreover, the panel was not designed to detect CNVs or gene rearrangements, although this limitation was resolved in part by analyzing the genes most frequently amplified in BC by FISH.

Further research in this field would require studies with a larger number of patients and samples and perhaps focused on patients with BC of the TN surrogated molecular type, as this seems to be not only the most likely to metastasize to the skin but also has the poorest prognosis.

## 5. Conclusions

The development of new strategies for the management of CMs is a major clinical challenge because of the poor prognosis. To advance in this field, a better understanding of the molecular alterations involved in the metastatic process is needed. In the present study, the clinicopathological characteristics of BCs developing CM was analyzed and compared to the molecular differences between primary breast tumors and their corresponding CMs. We observed that the surrogate molecular type of BC with a greater risk to metastasize to skin was TN. A change of tumor surrogate molecular type in metastases with an impact on treatment was found in 15% of patients. In addition, half of the CM presented some additional molecular alterations with respect to the primary tumors, but a characteristic molecular pattern related to tumor progression and CM development was not observed. In this series, survival was related to the tumor surrogate molecular type. The immunohistochemical and molecular analysis of BC CM is essential for a proper treatment of the patients.

## Figures and Tables

**Figure 1 cancers-14-01151-f001:**
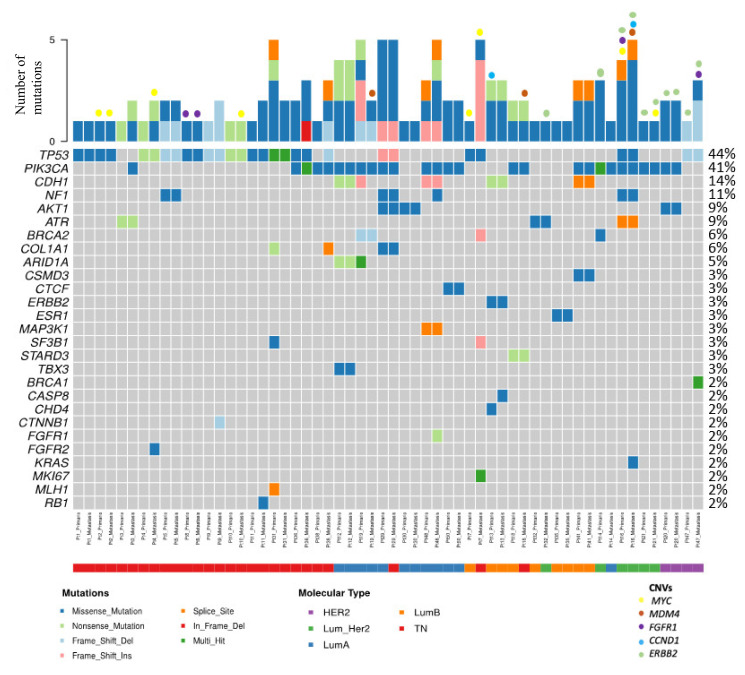
Distribution of mutations and CNVs in this series. The figure shows paired samples corresponding to 29 paired patients, but the mutation frequencies were calculated considering the 33 paired cases with 66 samples.

**Figure 2 cancers-14-01151-f002:**
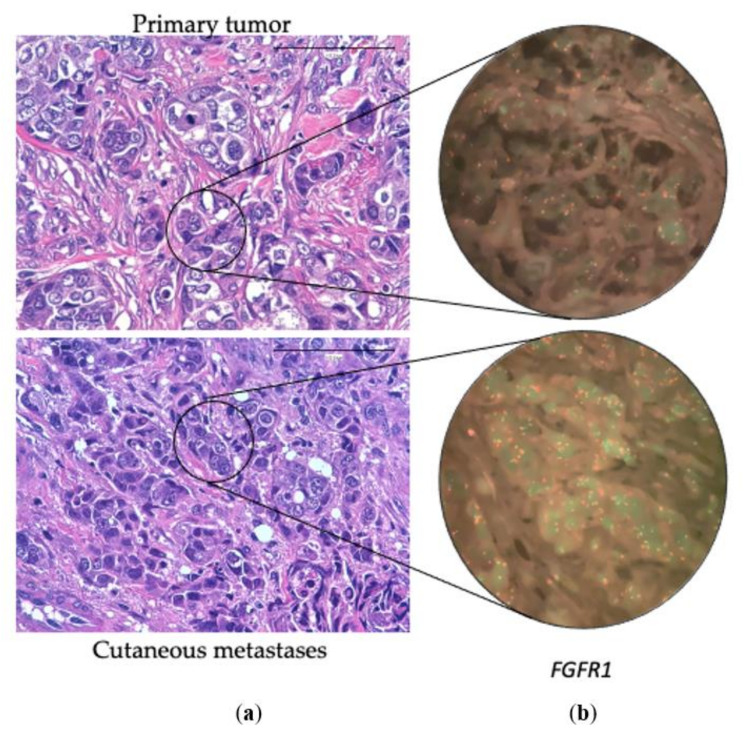
(**a**) Hematoxylin-eosin of a primary breast tumor and its corresponding cutaneous metastasis. (**b**) Fluorescent In-Situ Hybridization of the FGFR1 gene in the primary breast tumor (without CNV) and in the cutaneous metastasis (with polysomy). 100×.

**Figure 3 cancers-14-01151-f003:**
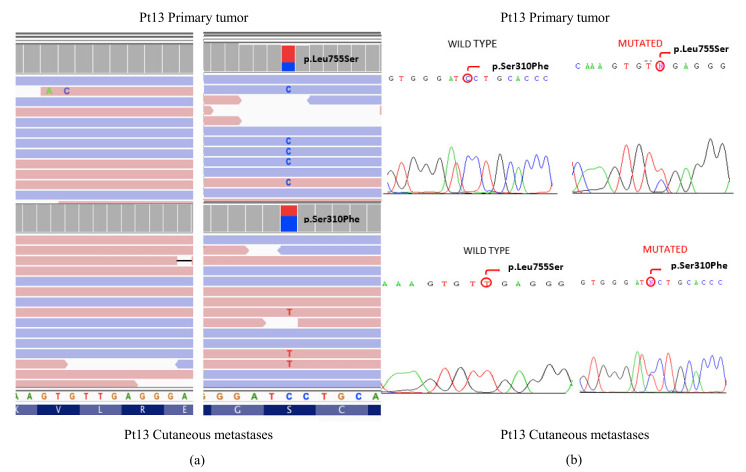
(**a**) Visualization in the IGV software of different *ERBB2* mutations found in the primary tumor and the cutaneous metastasis in patient Pt13. (**b**) Orthogonal validation by the Sanger sequencing.

**Figure 4 cancers-14-01151-f004:**
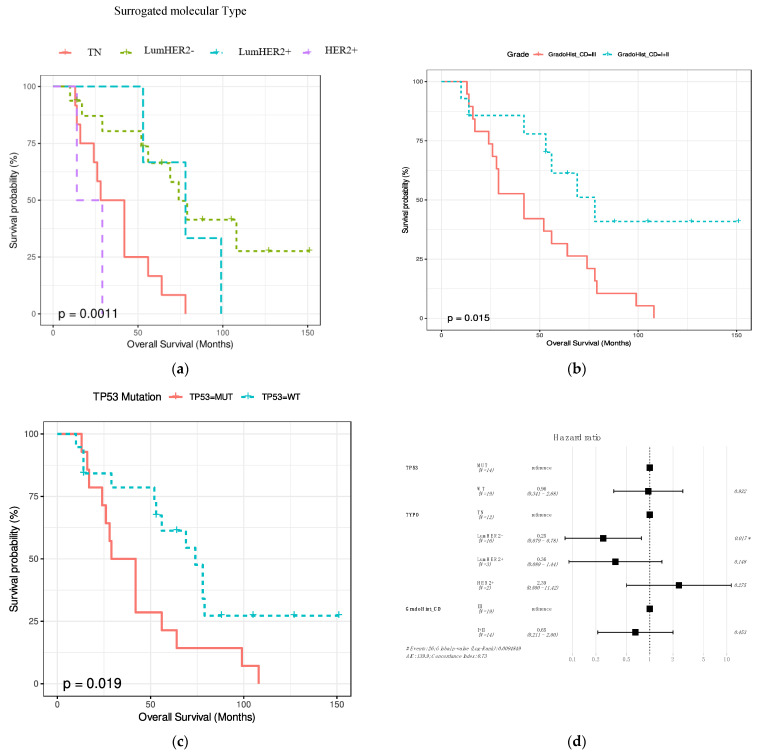
Kaplan–Meyers graphs showing the association between overall survival and the surrogate molecular type (**a**), histological grade (**b**), and *TP53* status (**c**). (**d**) Multivariate analysis showing the independent prognostic significance of the surrogate molecular type.

**Table 1 cancers-14-01151-t001:** Clinicopathological features of the 33 patients.

Clinicopathological Features	Categories		*n* (%)
Sex	Female		33 (100)
Age	<60		14 (42.4)
	>60		19 (57.6)
Cutaneous metastases location	Local *		15 (46.9)
	Distance		17 (53.1)
	NA		1
Menopausal status at diagnosis	Yes		27 (90)
	No		3 (10)
	NA		3
pT	1		5 (18.5)
	2		14 (51.9)
	3		5 (18.5)
	4		3 (11.1)
	NA		6 **
pN	0		8 (30.8)
	1		10 (38.5)
	2		2 (7.7)
	3		6 (23)
	NA		7 **
Clinical stage	I		4 (12.5)
	II		5 (15.6)
	III		14 (43.7)
	IV		9 (28.1)
	NA		1
Histological Grade	1		1 (3)
	2		13 (39.4)
	3		19 (57.6)
LIV	Yes		11 (33.3)
	No		22 (66.7)
Immunohistochemical markers	ER+		19 (57.6)
	PR+		11 (33.3)
	HER2+		5 (15.2)
	Ki67	≤15%	9 (27.3)
		16–29%	5 (15.2)
		≥30%	19 (57.6)
	TN		12 (36.4)
Surrogated molecular type	Luminal HER2-		16 (48.5)
	Luminal HER2+		3 (9)
	HER2+ (non-luminal)		2 (6)

* Local lesions refer to lesions that presented on the skin of the breast/thorax. ** Patients diagnosed at stage IV did not undergo surgery and did not have pT and pN data.

**Table 2 cancers-14-01151-t002:** Surrogate molecular type change between primary tumors and their cutaneous metastasis.

Tumor	Surrogated Molecular Type
Pt7_Primary	Luminal HER2-
Pt7_Metastasis	TN
Pt18_Primary	Luminal HER2-
Pt18_Metastasis	TN
Pt29_Primary	Luminal HER2-
Pt29_Metastasis	TN
Pt14_Primary	Luminal HER2+
Pt14_Metastasis	Luminal HER2-
Pt32_Primary	Luminal HER2-
Pt32_Metastasis	Luminal HER2+

**Table 3 cancers-14-01151-t003:** Pathogenic alterations in primary tumors and their cutaneous metastasis.

Surrogate Molecular Types		Gene	Primary Tumors *n* (%)	Cutaneous Metastases *n* (%)
TN *n =* 12	Mutations	*TP53*	10 (83)	11 (92)
		*PIK3CA*	2 (17)	4 (33)
		*NF1*	1 (8.3)	1 (8.3)
TN *n =* 7	CNVs	*MYC*	1 (14.3)	3 (42.9)
		*MDM4*	0	1 (14.3)
		*FGFR1*	1 (14.3)	1 (14.3)
Luminal HER2- *n =* 16	Mutations	*TP53*	1 (6)	1 (6)
		*PIK3CA*	7 (39)	7 (39)
		*NF1*	1 (6.2)	2 (12.5)
		*AKT1*	2 (12.5)	2 (12.5)
Luminal HER2- *n =* 9	CNVs	*MYC*	1 (11.1)	1 (11.1)
		*MDM4*	0	1 (11.1)
		*CCND1*	1 (11.1)	0
HER2+ *n =* 5	Mutations	*TP53*	2 (40)	2 (40)
		*PIK3CA*	4 (80)	4 (80)
		*NF1*	1 (20)	1 (20)
		*AKT1*	1 (20)	1 (20)
HER2+ *n =* 4	CNVs	*MYC*	1 (25)	1 (25)
		*MDM4*	0	1 (25)
		*FGFR1*	1 (25)	1 (25)
		*CCND1*	0	1 (25)

**Table 4 cancers-14-01151-t004:** Additional molecular alteration in cutaneous metastases not found in primary tumors of paired cases.

Location	Surrogated Molecular Type	Gene	Cases with Additional Mutation in Cutaneous Metastasis *n*
Distant cutaneous metastasis *n* = 17	TN	*PIK3CA*	1
		*RB1*	1
		*FGFR2 + MYC* (amplification)	1
		*CTNNB1*	1
		*MDM4* (amplification)	1
	Luminal HER2-	*FGFR1 + NF1*	1
		*BRCA2 + MKI67 + SF3B1*	1
		*CASP8 + ERBB2*	1
		*MDM4* (amplification)	1
	HER2+	*KRAS + CCND1* (amplification) + *MDM4* (amplification)	1
		*BRCA1 + FGFR1* (polysomy)	1
Local cutaneous metastasis *n* = 15	TN	*TP53 + PIK3CA + COL1A1*	1
		*PIK3CA*	1
		*MYC* (amplification)	1
	Luminal HER2-	*ERBB2* (polysomy)	1
	HER2+	*MYC* (amplification)	1

When not specified as an amplification, the alteration detected was a mutation.

**Table 5 cancers-14-01151-t005:** Distribution of surrogated molecular types in breast cancer with cutaneous metastasis in different series.

Authors	*n*	Luminal HER2− *n* (%)	HER2+ *n* (%)	TN *n* (%)	Unknown *n* (%)
Yates y col. [39]	19	9 (47)	2 (10)	5 (26)	3 (16)
Kong y col. [15]	125	53 (42.4)	43 (34)	29 (23)	
Luna y col. [38]	26	7 (27)	7 (27)	10 (39)	2 (7)
González-Martínez y col. [37]	58	29 (50)	8 (14)	15 (26)	6 (10)
Present series	33	16 (48.5)	5 (15.2)	12 (36.4)	

**Table 6 cancers-14-01151-t006:** Additional mutations in cutaneous metastases reported in different series.

Authors	Paired Cases of Cutaneous Metastases *n*	Cases with Additional Mutation	Additional Molecular Alterations in Cutaneous Metastases	
Schrijver and col. [41]	8 *	6	33 mutations (*ATR*, *BRCA1*, *SMAD4*, *CDH1*, *ARID1A*, *ERBB2*, *IDH1*, *PIK3R1*, *RB1*, etc.) and *FGF3* amplification	
Yates and col. [39]	Cohort 1: 2	2	4 molecular alterations (*FGFR1* amplification/*TP53* structural variant, *RB1* indel/*TERC* amplification)	
	Cohort 2: 4	4	8 mutations (*JAK2*, *NF1*, *TP53*, *AKT1*, *ARID1A, ARID1A, RB1*) and 2 amplifications (*MYC* and *FGFR1*)	
Paul and col. [42]	1	1	54 mutations (*PIK3CA*, *TP53*, etc.)	
Present series	33		12 TN	6 mutations *(**TP53 + PIK3CA + COL1A1, PIK3CA, RB1, FGFR2, CTNNB1)* and 3 amplifications (2 in *MYC* and one in *MDM4)*
		17	16 RH + HER2-	7 mutations *(FGFR1 + NF1, BRCA2 + MKI67 + SF3B1, CASP8+ ERBB2*) and 2 amplifications or polysomy (*ERBB*2 and *MDM4*)
			5 HER2+	2 mutations (*KRAS, BRCA1*), 4 amplifications or polysomy (*CCND1*, *MYC, FGFR1* and *MDM4*)

* Of these 8 cases, 2 had no additional molecular alterations in cutaneous metastasis.

## Data Availability

The data presented in this study are available on request from the corresponding author.

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
