# Peer review of "Differences in the Molecular Profile between Primary Breast Carcinomas and Their Cutaneous Metastases"

_cancers, 2022, doi:10.3390/cancers14051151_

Round 1

Reviewer 1 Report

The answer to the specific question raised in the title of the article is that there is no relationship between the molecular profile and cutaneous metastases that would allow a causal relationship to be established that explains cutaneous progression.

Given that all diagnostic stages are included in the 33 women with breast cancer, what has not been taken into account and should at least be commented on, is the impact of the different treatments on the differences in the molecular profile. 

I think a comment should be made about it.

Reviewer 2 Report

The manuscript confirms that triple negative (BCTN) is the breast cancer molecular subtype with a greater risk to metastasize to skin.

The authors added information testing a panel of 61 genes and, comparing the primary tumors with their matched cutaneous metastases (CM), helped to conclude that half of the CM presented some additional molecular alterations with respect to the primary tumors. Nevertheless, a characteristic molecular pattern related to tumor progression and CM development was not possible to detect. In general, these findings may help to take treatment decisions; on the other hand, confirm previous reports which sustained no specific pattern of mutations that predispose to CM.

Reviewer 3 Report

The Authors provide a well described analysis of primary tumors and cutaneous metastasis (CM) in BC. The y found thattTriple negative (TN) BCs were overrepresented  among tumors that developed CM. 48.5% of the CM presented some additional molecular alteration with respect to the primary tumor. Survival was related to histological grade, tumor surrogate molecular type and TP53 mutations in the univariate analysis but only the tumor surrogate molecular type remained as a prognostic factor in the multivariate analysis. The Authors concluded that the TN molecular type has a greater risk to develop skin metastasis., and there are were phenotypic changes and additional molecular alterations in skin metastases compared to the corresponding primary breast tumors in nearly half of the patients. All these data provide evidnece tha tthese alteration metastasis-related may have a impact on treatment response.

This analysis, although based on a relative small number of cases, is well defined and evaluated , the statistical methods are sound and NGS data were confirmed with an independent technique (Sanger). I have no additional comments to ask.
